# Serological responses and vaccine effectiveness for extended COVID-19 vaccine schedules in England

Gayatri Amirthalingam [1,8 ✉], Jamie Lopez Bernal[1,8], Nick J. Andrews[1], Heather Whitaker [2], Charlotte Gower[1], Julia Stowe[1], Elise Tessier [1], Sathyavani Subbarao[1], Georgina Ireland[1], Frances Baawuah[1,3], Ezra Linley[4], Lenesha Warrener[5], Michelle O'Brien[3], Corinne Whillock[1], Paul Moss [6], Shamez N. Ladhani[1,7], Kevin E. Brown [1,9] & Mary E. Ramsay[1,9]

The UK prioritised delivery of the first dose of BNT162b2 (Pfizer/BioNTech) and AZD1222 (AstraZeneca) vaccines by extending the interval between doses up to 12 weeks. In 750 participants aged 50–89 years, we here compare serological responses after BNT162b2 and AZD1222 vaccination with varying dose intervals, and evaluate these against real-world national vaccine effectiveness (VE) estimates against COVID-19 in England. We show that antibody levels 14–35 days after dose two are higher in BNT162b2 recipients with an extended vaccine interval (65–84 days) compared with those vaccinated with a standard (19–29 days) interval. Following the extended schedule, antibody levels were 6-fold higher at 14–35 days post dose 2 for BNT162b2 than AZD1222. For both vaccines, VE was higher across all age-groups from 14 days after dose two compared to one dose, but the magnitude varied with dose interval. Higher dose two VE was observed with >6 week interval between BNT162b2 doses compared to the standard schedule. Our findings suggest higher effectiveness against infection using an extended vaccine schedule. Given global vaccine constraints these results are relevant to policymakers.

[1] Immunisation and Vaccine Preventable Diseases Division, UK Health Security Agency, London, United Kingdom. [2] Statistics, Modelling and Economics Department, UK Health Security Agency, London, United Kingdom. [3] Brondesbury Medical Centre, Kilburn, London, United Kingdom. [4] Sero-Epidemiolgy Unit, UK Health Security Agency, Manchester, United Kingdom. [5] Virus Reference Department, UK Health Security Agency, London, United Kingdom. [6] Institute of Immunology and Immunotherapy, University of Birmingham, Edgbaston, United Kingdom. [7] Paediatric Infectious Diseases Research Group, St. George's University of London, London, United Kingdom. [8] These authors contributed equally: Gayatri Amirthalingam, Jamie Lopez Bernal. [9] These authors jointly supervised this work: Kevin E. Brown, Mary E. Ramsay. ✉email: gayatri.amirthalingam@phe.gov.uk

Older adults have been disproportionately affected by the COVID-19 pandemic, with age being the single most important risk factor for hospitalisations and deaths[1–3]. In the United Kingdom, older adults were prioritised for vaccination at the start of the COVID-19 immunisation programme on 08 December 2020, initially with the Pfizer/BioNTech (BNT162b2) vaccine using the authorised 3-week interval between doses[4]. From 04/01/2021, the AstraZeneca (AZ) vaccine (AZD1222) was deployed and, with its more favourable storage and transport conditions, was used for vaccinating in care homes, community healthcare professionals and healthy adults aged 40–60 years. In January 2021, the UK Joint Committee on Vaccination and Immunisation (JCVI) recommended that with the emergence of a second wave as a result of the Alpha variant, the second dose of vaccine should be extended for up to 12 weeks to prioritise the first dose for those at highest risk of severe COVID-19 and death[5]. The decision to extend the second dose was based on early clinical trial data indicating nearly 90% effectiveness against SARS-CoV-2 within 3 weeks of the first dose of BNT162b2 vaccine compared to 95% from two weeks after the second dose[6]. Vaccinating more at-risk individuals quickly with a single dose was predicted to prevent more cases, hospitalisations and deaths than two doses at a 3-week interval[7]. This unique approach against authorised use and without formal clinical trials resulted in considerable international debate and prompted the need to evaluate immune responses and vaccine effectiveness following extended schedules.

The COVID-19 vaccine responses after extended immunisation schedules (CONSENUS) evaluation aimed to assess immune responses in ≥50 year-olds receiving a COVID-19 vaccine as part of the UK extended immunisation schedule. Early analysis indicated that a single dose of BNT162b2 vaccine was associated with >94% seropositivity after 3 weeks in previously uninfected older adults, while two doses produced very high antibody levels, significantly higher than convalescent sera from adults with mild-to-moderate PCR-confirmed COVID-19[5]. Real-world effectiveness studies indicate 50–70% protection against infection or mild disease for ≥8 weeks after one BNT162b2 dose and ≥6 weeks after AZD1222, with 75–85% protection against hospitalisation or death[8].

We now report serological responses in 750 adults aged 50–89 years given two doses of BNT162b2 or AZD1222 at different intervals, comparing serological responses. These findings are evaluated against real-world vaccine effectiveness estimates against COVID-19 using similar dosing intervals in the same age group in England.

## Results

**Participants**. We recruited 750 participants aged 50–89 years (median age, 71, IQR 66–76 years)- 421 received at least one BNT162b2 dose and 329 at least one AZD1222 dose (Table 1 and Supplementary Table 1). Overall, 46% (344/746) were male, 27% (171/743) were of non-White ethnicity, 16.8% (126/750) had evidence of the previous infection at enrolment and one seroconverted during the study. Adults aged 50–64 years were more likely to have evidence of previous infection than older adults (56/171; 32.8% vs 70/579; 12.1%; $X^2(1) = 40.3$, $p < 0.001$).

**Post-dose 1: Antibody responses in uninfected adults**. Among BNT162b2 vaccine recipients receiving an extended schedule, seropositivity increased rapidly after the first dose, with 97.7% (217/222) seroconverting by 17–34 days and 35–55 days (97.7%; 254/260) post-vaccination. S-antibody GMTs peaked by 35–55 days after vaccination at 29.8 (95%CI: 24.9–35.6) for 64–79

year-olds and 41.7 (95%CI: 28.5–60.8) for 80–89 year-olds, with levels sustained to 77–97 days post-dose 1 (Fig. 1 and Table 2).

Among AZD1222 vaccine recipients, 85.9% (55/64) of 50–64 year-olds and 81.7% (49/60) of 65–79 year-olds seroconverted at 17–24 days, which increased to 95.6% (198/207) overall at 35–55 days. GMTs continued to increase from 13.7 (95%CI: 7.5–24.9) at 17–34 days to 38.2 (95%CI: 24.9–58.7) at 35–55 days in 50–64 year-olds, whilst, in older adults, the GMT peak was delayed until 56–76 days. The GMR for S-antibody was 62.01 (95%CI: 47.69–80.64) for BNT162b2 vaccine recipients aged 65–79 years at 17–34 days compared to pre-vaccine, followed by a GMR of 1.2 (95%CI: 1.08–1.34) from 17–34 days to 35–55 days after dose 1. For the AZD1222 vaccine in the same age group, GMRs were 14.68 (95%CI: 9.68–22.26) and 4.17 (95%CI: 3.28–5.3), respectively. In previously-uninfected individuals, GMTs remained lower in the 13 weeks post-dose 1 for both vaccines compared to convalescent sera from mild-to-moderate PCR-confirmed cases (Fig. 1 and Table 1).

GMRs between BNT162b2:AZD1222 in previously-uninfected recipients was 4.05 (95%CI: 2.49–6.6) at 17–34 days after dose 1, this ratio declined with time and was no longer statistically significant at 56–76 days post-vaccination (Table 2 and Fig. 1). Females had higher S-antibody GMTs than males, while differences in age groups narrowed with time since vaccination such that there was no significant difference by age group at 56–76 days post-vaccination (Table 2 and Fig. 1).

**Post-dose 1: Vaccinees with previous COVID-19 infection**. In adults with serological evidence of prior infection, S-antibody levels at vaccination were not significantly different from convalescent sera at 56–98 days post-infection (Table 2). S-antibody GMTs increased from 71.4 (95%CI: 12.0–424.5) to 3,842.9 (95%CI: 1229.4–12012.4) at 17–34 days post-vaccination for BNT162b2 recipients, and from 127.8 (95%CI: 75.9–215.1) to 12616.8 (95%CI: 8880.7–17924.8) for AZD1222 vaccine recipients. These initially high titres subsequently waned through 56–76 days after dose 1.

**Post-dose 1: Vaccine effectiveness sustained ≥8 weeks following dose 1**. The odds of testing SARS-CoV-2 PCR-positive among vaccinated people increased up to days 7–9 after dose 1 for both vaccines, reaching 1.12 (VE: −12%) and 1.19 (VE: −19%) for AZD1222 and BNT162b2 in 65–79 year-olds, respectively (Fig. 2 and Supplementary Table 2). Among ≥80 year-olds receiving BNT162b2, VE increased from days 14–20, reaching 61% (95% CI: 49–71) in the early cohort (three-week interval) and 52% (95%CI: 39–63) in the later (longer interval) cohort at 28–34 days and remained at similar levels between days 35–55 (5–8 weeks). Amongst 65–79 year-olds, VE began to increase from 10–13 days after vaccination, reaching 53% (95%CI: 45–60) on days 28–34, and remained at a similar level between days 35–69 (5–10 weeks). A similar trend was observed in the BNT162b2 recipients aged 50–64 years with a VE of 58% at days 28–34. Whilst there was some evidence of a 10–20% decrease in VE by 10 weeks after the first dose, there was an apparent rise again in VE at the final interval, although with wide confidence intervals (Fig. 2).

For the AZD1222 vaccine, the positive VE within 3 days of vaccination was likely an artefact because vaccinated adults were getting PCR-tested and reported test-negative due to vaccine reactogenicity (Fig. 2 and Supplementary Table 3). In adults aged ≥80 years, VE increased from days 14–20, reaching 43% (95%CI: 24–58) on days 28–34 and remained at a similar level between days 35–55 (5–8 weeks). Amongst 65–79 year-olds, VE increased from days 14–20 post-vaccination, reaching 55% (95%CI: 48–61) after 28 days, and then remaining stable until days 56–69 after the

**Table 1 Proportion RocheS seropositive, Geometric Mean Concentrations (if >5 samples) by vaccine type and age group and the ratio of responses by time, following dose 1.**

| | Vaccine and age group | Time after dose 1 | N | positive (%) | Geometric mean (95% CI) | Ratio of response from time window prior (95% CI) |
|---|---|---|---|---|---|---|
| Previously uninfected | AstraZeneca, extended schedule, ages 50-64 | 0 | 51 | 0 (0%) | 0.4 (0.4-0.4) | |
| | | 17-34 | 64 | 55 (85.9%) | 13.7 (7.5-24.9) | 30.92 (14.89-64.22) |
| | | 35-55 | 89 | 86 (96.6%) | 38.2 (24.9-58.7) | 2.78 (2.12-3.64) |
| | | 56-76 | 44 | 42 (95.5%) | 38.6 (20.5-72.8) | 0.99 (0.73-1.36) |
| | | 77-97 | 5 | 5 (100%) | | |
| | AstraZeneca, extended schedule, ages 65-79 | 0 | 72 | 0 (0%) | 0.4 (0.4-0.4) | |
| | | 17-34 | 60 | 49 (81.7%) | 5.9 (3.7-9.5) | 14.68 (9.68-22.26) |
| | | 35-55 | 110 | 105 (95.5%) | 22.9 (17.5-30) | 4.17 (3.28-5.3) |
| | | 56-76 | 60 | 57 (95%) | 30.1 (20.6-44) | 1.48 (1.16-1.88) |
| | | 77-97 | 11 | 10 (90.9%) | 17.6 (6.5-47.8) | 1.02 (0.61-1.69) |
| | AstraZeneca, extended schedule, ages 80-89 | 0 | 5 | 0 (0%) | | |
| | | 17-34 | 8 | 6 (75%) | 4.6 (0.9-22.9) | 11.46 (2.14-61.45) |
| | | 35-55 | 8 | 7 (87.5%) | 16.2 (2.8-94.5) | 2.63 (1.24-5.58) |
| | | 56-76 | 6 | 5 (83.3%) | 12.3 (1-157.8) | 1.05 (0.51-2.16) |
| | Pfizer, extended schedule, ages 65-79 | 0 | 115 | 0 (0%) | 0.4 (0.4-0.4) | |
| | | 17-34 | 141 | 138 (97.9%) | 25.2 (19.8-32) | 62.01 (47.69-80.64) |
| | | 35-55 | 206 | 201 (97.6%) | 29.8 (24.9-35.6) | 1.2 (1.08-1.34) |
| | | 56-76 | 165 | 161 (97.6%) | 26.1 (21.6-31.6) | 0.89 (0.8-0.98) |
| | | 77-97 | 6 | 6 (100%) | 36.9 (10.2-132.9) | 1.07 (0.68-1.68) |
| | Pfizer, extended schedule, ages 80-89 | 0 | 1 | 0 (0%) | | |
| | | 17-34 | 58 | 56 (96.6%) | 24 (15.2-37.9) | |
| | | 35-55 | 54 | 53 (98.1%) | 41.7 (28.5-60.8) | 1.45 (1.23-1.71) |
| | | 56-76 | 54 | 54 (100%) | 38.1 (27.6-52.7) | 0.88 (0.74-1.03) |
| | | 77-97 | 8 | 8 (100%) | 17.8 (5-63.2) | 0.89 (0.62-1.27) |
| Previously infected | AstraZeneca, extended schedule, all ages | 0 | 28 | 28 (100%) | 127.8 (75.9-215.1) | |
| | | 17-34 | 37 | 37 (100%) | 12616.8 (8880.7-17924.8) | 87.73 (56.47-136.28) |
| | | 35-55 | 64 | 64 (100%) | 10621.3 (7749.1-14558) | 0.83 (0.75-0.93) |
| | | 56-76 | 42 | 42 (100%) | 6984.4 (4749.8-10270.3) | 0.76 (0.7-0.84) |
| | | 77-97 | 8 | 8 (100%) | 5599.7 (1617.7-19383.8) | 0.82 (0.68-0.99) |
| | Pfizer, extended schedule, all ages | 0 | 12 | 10 (83.3%) | 71.4 (12-424.5) | |
| | | 17-34 | 19 | 19 (100%) | 3842.9 (1229.4-12012.4) | 54.88 (26.79-112.41) |
| | | 35-55 | 32 | 32 (100%) | 5522.5 (2901.6-10510.5) | 0.82 (0.68-0.99) |
| | | 56-76 | 25 | 25 (100%) | 2853.2 (1447-5626.2) | 0.82 (0.69-0.98) |
| | Both vaccines | 0 | 40 | 38 (95%) | 107.3 (58.8-195.9) | |
| Convalescent sera, by days post symptom onset | Unvaccinated, ages 50-89 | 35-55 | 141 | 134 (95%) | 55.3 (39.4, 77.7) | |
| | | 56-98 | 87 | 86 (98.9%) | 128.2 (89.2, 184.3) | |

first vaccine dose. A similar trend was observed among 50–64 year-olds, with 50% (95%CI: 45–55) VE estimates. A reduction in VE was noted by 70 days post-vaccination at 40% (95%CI: 23–53) and 26% (95%CI: 18–33) for 65–79 year-olds and 50–64 year-olds, respectively. For ≥80 year-olds, confidence intervals were too wide to asses declines, but point estimates showed a decline after 8 weeks.

**Post-dose 2: Antibody responses by dosing interval.** Vaccine dose intervals varied between 3–13 weeks amongst CONSENSUS participants. They were, therefore, assessed by the following intervals: (i) 19–29, (ii) 45–64, (iii) 65–84, and (iv) ≥85 days. Sampling timepoints post-dose 2 were divided into 7–13 and 14–34 days. The vaccine used and dose intervals varied by age group depending on national recommendations and vaccine supply. Those receiving BNT162b2 at 19–29 day intervals and those reaching 85+ days for either vaccine were older (median age 76, 76 and 74 years, respectively) whilst those receiving

AZD1222 at 45–65 and 65–84 intervals were younger (66 and 66.5 years, respectively) because initial vaccine rollout prioritised older adults and recommended a 3-week interval between doses.

Regardless of vaccine and schedule, all participants seroconverted 7+ days after dose 2 (AZD1222: $N = 200$; BNT162b2 extended: $N = 282$ and BNT162b2 control: $N = 87$). BNT162b2 dose 2 responses were quick, peaking at 7–13 days followed by a 23% decline at 14–34 days. Amongst BNT162b2 recipients, GMTs were tenfold higher at 14–34 days post-dose 2 following a 65–84 day interval compared with a 19–29 day interval (Table 3 and Fig. 3). Furthermore, among those with a vaccine interval of 85+ days, GMTs at 7–13 days post-dose 2 were higher compared with 65–84 days. There were, however, too few results to confirm this trend beyond 14 days post-dose 2.

GMTs with a 65–84 day interval were sixfold higher after BNT162b2 (6703; 95%CI, 5887–7633) compared to AZD1222 (1093; 95%CI: 806–1483) at 14–34 days post-dose 2 (Table 4). However, GMTs among AZD1222 recipients with an extended schedule were significantly higher than those receiving the shorter

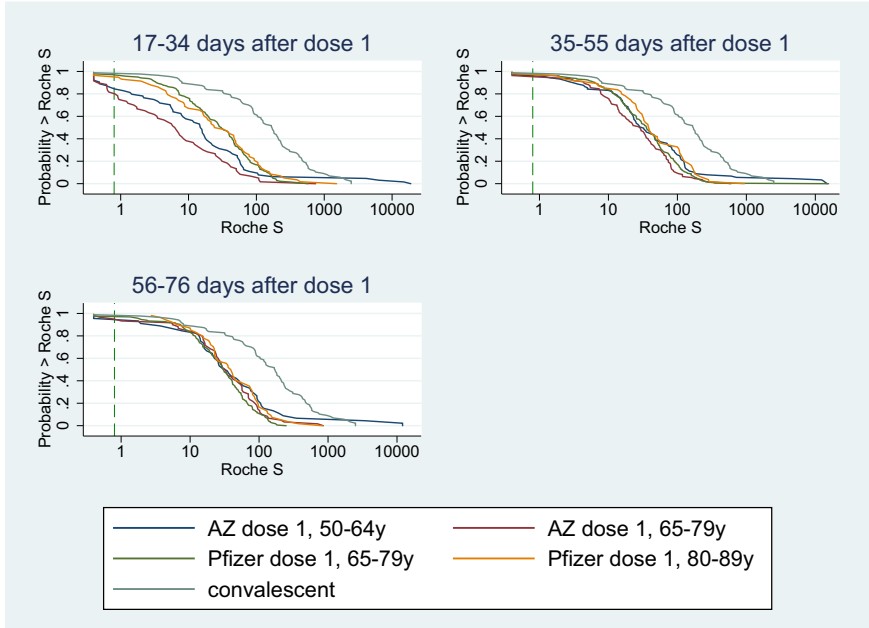

**Fig. 1 Reverse cumulative distribution curves, antibody responses following the first dose of COVID-19 vaccine in previously uninfected individuals, by the vaccine, age group and including a curve for unvaccinated convalescent cases 56–98 days post-infection.** S-antibody levels at 17–34, 35–55 and 56–76 days after dose 1. Blue: AstraZeneca (AZ) recipients aged 50–64 years old; Red: Az recipients aged 65–79 years old; Green: Pfizer recipients aged 65–79 years old; Yellow: Pfizer recipients aged 80–89 years old; Light blue/grey: Post SARS-CoV-2 infection convalescent plasma. The green dashed line is the assay positive cut-off (0.8).

**Table 2 Geometric mean ratio (GMR) of responses, adjusted for age and sex, following dose 1 of an extended vaccine schedule. *P* values relate to two-sided *z*-tests of log(GMR) = 0.**

| | | 0 weeks | | 17–34 days | | 35–55 days | | 56–76 days | |
|---|---|---|---|---|---|---|---|---|---|
| | | Geometric mean ratio of (log) RocheS responses | *p* value | Geometric mean ratio of (log) RocheS responses | *p* value | Geometric mean ratio of (log) RocheS responses | *p* value | Geometric mean ratio of (log) RocheS responses | *p* value |
| vaccine | AstraZeneca | 1(ref) | | 1(ref) | | 1(ref) | | 1(ref) | |
| | Pfizer | 0.99 (0.98–1) | 0.171 | 4.05 (2.49–6.6) | <0.001 | 1.39 (1–1.93) | 0.049 | 1 (0.68–1.48) | 0.987 |
| age group | 50–64 | 1(ref) | | 1(ref) | | 1(ref) | | 1(ref) | |
| | 65–79 | 1 (0.99–1.01) | 0.928 | 0.49 (0.27–0.9) | 0.021 | 0.58 (0.38–0.87) | 0.009 | 0.68 (0.4–1.15) | 0.151 |
| | 80–89 | 0.99 (0.97–1.02) | 0.692 | 0.47 (0.22–0.98) | 0.045 | 0.69 (0.39–1.21) | 0.196 | 0.8 (0.42–1.52) | 0.492 |
| sex | male | 1(ref) | | 1(ref) | | 1(ref) | | 1(ref) | |
| | female | 1 (0.99–1.01) | 0.481 | 1.24 (0.84–1.82) | 0.273 | 1.85 (1.41–2.43) | <0.001 | 1.72 (1.26–2.34) | 0.001 |

(19–29 days) BNT162b2 schedule (694; 95%CI: 540–893). Unlike BNT162b2 recipients, there was no decline in antibody titres among AZD1222 recipients between 7–13 and 14–34 days post-dose 2 regardless of interval. Responses were twofold higher among AZD1222 recipients with a 65–84 compared with 45–64 day dose interval. Responses were, however, lower following an 85+ day interval between AZD1222 doses, although this group was small, with older participants and lower dose 1 responses.

In all groups, GMTs after two vaccine doses regardless of the interval was higher than those observed after mild-to-moderate COVID-19 (Table 3 and Fig. 3).

In participants previously infected, following dose 2 antibodies were further boosted by BNT162b2, increasing to 27322.5 (95% CI: 17444.4–42794.2), but not by AZD1222, where the GMT was 9633.2 (95%CI: 6233.9–14886.3) at 14–34 days post-dose 2.

**Post-dose 2: Vaccine effectiveness following extended schedules.** VE was higher across all age groups for both vaccines from 14 days after dose 2 compared to dose 1 but the magnitude

depended on the interval between doses (Fig. 4 and Supplementary Tables 2, 3). Amongst BNT162b2 recipients, VE was consistently higher with > 45-day intervals compared to 19–29 days for all age groups.

Amongst AZD1222 recipients aged ≥80 years, two-dose VE after 14 days was 96% (95%CI: 68–99) and 82% (95%CI: 68–89) following 45–64 and 65–84 days intervals, respectively (Fig. 4). Those receiving their second dose outside of these recommended intervals also had high VE after two doses; for ≥ 85-day intervals, the estimated VE was 88% (95%CI: 48–97). In younger adults, two-dose VE was higher but not statistically significant with a 65–84 than a 45–64 day interval, but lower at all timepoints than ≥80 year-olds (Fig. 4 and Table Supplementary files).

**Discussion**

Our findings uniquely combine serological and vaccine effectiveness data for extended immunisation schedules in adults who were prioritised for vaccination during the first phase of the UK COVID-19 immunisation programme. We demonstrate high and sustained antibodies responses for >12 weeks after the first dose

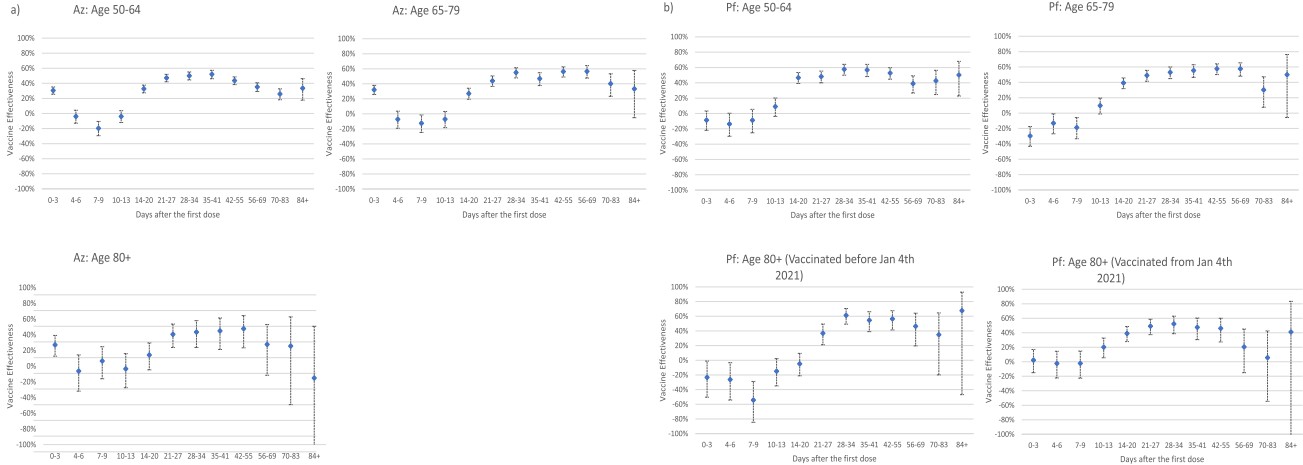

**Fig. 2 Adjusted vaccine effectiveness against confirmed symptomatic COVID-19 by the interval after vaccination amongst 50–64 year-olds, 65–79 year-olds and 80+ year-olds with the (a) AZ vaccine and (b) Pfizer-BioNTech BNT162b2 vaccine. a** VE against confirmed symptomatic COVID-19 after vaccination with AstraZeneca. $N = 425,907$ people PCR tested. Logistic regression was used to estimate the odds of vaccination in PCR-confirmed cases compared with those who tested negative for SARS-CoV-2. Vaccine effectiveness was calculated as 1 minus adjusted odds ratio and is presented as a percentage ±95% confidence intervals. **b** VE against confirmed symptomatic COVID-19 after vaccination with Pfizer. $N = 384,297$ people PCR tested. Logistic regression was used to estimate the odds of vaccination in PCR-confirmed cases compared with those who tested negative for SARS-CoV-2. Vaccine effectiveness was calculated as 1 minus adjusted odds ratio and is presented as a percentage ±95% confidence intervals.

**Table 3 Geometric mean responses and the geometric mean ratio of responses following dose 2, by the vaccine, interval compared with convalescent sera.**

| | Vaccine | Days between doses | Time since dose 2, days | N | Geometric mean response (95% CI)* | Geometric mean ratio of responses from timepoint prior (95% CI)* |
|---|---|---|---|---|---|---|
| Previously uninfected | AstraZeneca | 45–64 | 0–20 before | 58 | 29 (20–42) | |
| | | | 7–13 | 45 | 591 (474–738) | 19.11 (13.62–26.83) |
| | | | 14–34 | 42 | 583 (443–767) | 1.03 (0.89–1.19) |
| | | 65–84 | 0–2 before | 62 | 32 (23–44) | |
| | | | 7–13 | 53 | 857 (594–1238) | 26.5 (19.03–36.89) |
| | | | 14–34 | 60 | 1093 (806–1483) | 1.12 (0.91–1.39) |
| | | 85+ | 0–20 before | 12 | 17 (5–56) | |
| | | | 14–34 | 12 | 650 (206–2053) | |
| | Pfizer | 19–29 | 14–34 | 80 | 694 (540–893) | |
| | | 65–84 | 0–20 before | 197 | 29 (24–34) | |
| | | | 7–13 | 133 | 7198 (5820–8902) | 267.86 (229.48–312.66) |
| | | | 14–34 | 200 | 6703 (5887–7633) | 0.77 (0.71–0.83) |
| | | 85+ | 0–20 before | 9 | 32 (11–92) | |
| | | | 7–13 | 9 | 14437 (4136–50391) | 602.82 (416.34–872.82) |
| Previously infected | AstraZeneca | All | 0–20 before | 50 | 7458.3 (5417.9–10267.1) | |
| | | | 7–13 | 25 | 9138.2 (5997.4–13923.7) | 1.12 (0.94–1.33)* |
| | | | 14–34 | 26 | 9633.2 (6233.9–14886.3) | 0.93 (0.79–1.1) |
| | Pfizer | 19–29 | 14–34 | 7 | 17998.4 (4378.7–73982) | |
| | | All 30+ | 0–20 before | 25 | 2859.9 (1450.1–5640.5) | |
| | | | 7–13 | 18 | 40419 (28789–56747.2) | 14.49 (7.89–26.63)* |
| | | | 14–34 | 23 | 27322.5 (17444.4–42794.2) | 0.68 (0.54–0.85) |
| Convalescent sera in unvaccinated cases post onset | unvacc | | 21–55 | 141 | 55.3 (39.4, 77.7) | |
| | | | 56–90 | 87 | 128.2 (89.2, 184.3) | |

*GMRs at 7–13 days post two are relative to responses at 0–20 days before dose 2.

of either BNT162b2 or AZD1222 vaccine in previously-uninfected adults, with 97.7 and 95.6%, respectively, becoming seropositive by 35–55 days after their first dose. Antibody levels rose more rapidly and then stabilised after a single BNT162b2 dose but increase more gradually after AZD1222, such that antibody levels were equivalent in both cohorts by 56–76 days after a single dose. In previously-infected individuals, both

vaccines provided significant boosting after one dose, with S-antibody GMTs >50-fold higher than adults with mild-to-moderate COVID at 8–12 weeks post-infection. These serological findings are consistent with national surveillance data on clinical protection against symptomatic disease. VE after a single BNT162b2 dose was 53–58% after 28 days across all age groups, with no evidence of a decline in effectiveness with age and only a

modest decline in effectiveness beyond 56–76 days after the first dose. For AZD1222, single-dose VE was 43–55% beyond 28 days, with some evidence of a decline amongst the oldest age group beyond 10 weeks. The decision to prioritise the first dose of vaccine in the UK was made in the context of a second wave of infection where infection rates of hospitalisation were rapidly increasing, limited supply of vaccine and the emergence of a more transmissible Alpha variant (Alpha). Additionally, it was anticipated that delaying the second dose would enhance boosting, extending the longevity of protection following a second dose. Lengthening intervals between vaccine doses to enhance boosting is well-recognised from studies of other vaccines, including hepatitis B[9]. It is hypothesised that this provides more time for the maturation of the immune response, particularly of memory and B and T cells, following the priming dose, which enhances the effect of any additional doses and thereby potentially lengthens the duration of immunity. Enhanced

immunogenicity was demonstrated in the pre-licensure AZD1222 trial[10–12], but the lack of similar data for extended schedules using BNT162b2 prompted a rapid post-implementation evaluation to monitor serological responses and real-world effectiveness data for both COVID-19 vaccines in the UK. The potential risks of delaying second doses were leaving high-risk individuals incompletely protected with sub-optimal antibody responses allowing selection of new variants. Our findings, however, show sustained high levels of antibodies after the first dose to 12 weeks and is supported by the high one dose vaccine effectiveness estimates which were maintained till the second doses were given.

With BNT162b2, we found 8–10-fold higher GMTs after the second dose with a 6–9 or 10–13 week interval compared with the authorised 3-week interval, which was also associated with the more rapid waning of up to 50% between 1.5–3 weeks and 19 weeks after dose 2. These findings are consistent with our other as-yet unpublished study in ≥80 year-olds, where peak antibody responses were 3.5-fold higher following extended-schedule BNT162b2 immunisation although, interestingly, cellular responses were 3.6-fold lower with the extended interval schedule[13]. Whilst other studies have reported higher GMTs following vaccination, direct comparison is not possible given the titres have been reported in different units using different serological assays[14,15]. The current study, which includes a wider age range, found no evidence of a difference in antibody decline with age after either dose, although a recent study reported lower serum neutralisation and binding IgG/IgA after the first dose with increasing age and lower potency against SARS-CoV-2 variants of concern (VOC) among ≥80 year-olds[16]. Following the second dose, neutralisation against VOC was detectable regardless of age.

Among AZD1222, the initial clinical trials allowed permissive intervals between vaccine doses which showed minimal waning of antibodies or protection against symptomatic COVID-19 for up to 3 months after the first dose in healthy working-age adults, with better boosting after the second dose after a longer interval[17]. In our cohort, too, which focused on older adults, AZD1222 recipients had twofold higher antibodies after a 10–13 compared to 6–9 week interval.

In previously-infected adults, we observed significant boosting of antibody responses after the first dose of both vaccines and after the second dose of BNT162b2, but not AZD1222. Notably, though, antibody GMTs following two doses of either vaccine were higher than those observed following mild-to-moderate COVID-19 regardless of interval.

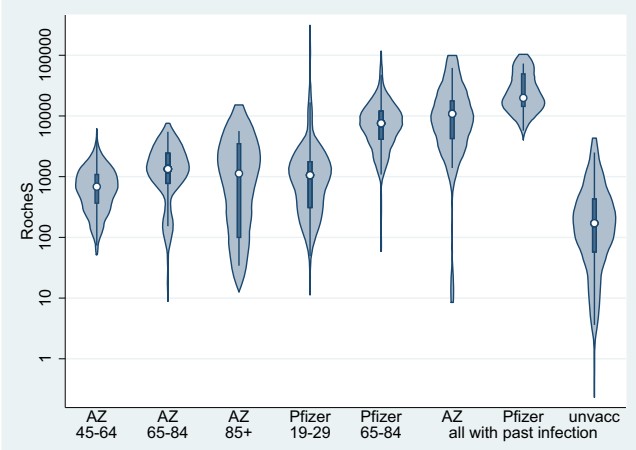

**Fig. 3 Violin plots, antibody responses at 14–34 days following the second dose of COVID-19 vaccine in (i) previously uninfected individuals, by vaccine and interval between doses, (ii) in previously infected individuals by the vaccine (any schedule) and (iii) in unvaccinated convalescent individuals. S-antibody response 14–34 days after dose 2**. N = 471 CONSENSUS participants tested 14–34 days after dose 2 and 87 convalescent sera in unvaccinated cases. The violin plots show the RocheS results of our CONSENSUS cohort. The centre represents the median, the thick central line the interquartile range, the thin central line the range (minus outliers) and the violin outline a smoothed kernel density estimation.

**Table 4 Adjusted models at two timepoints post-dose 2, using AZ 65–84 day schedule group as reference. P values relate to two-sided z-tests of log(GMR) = 0.**

|  |  | 7–13 days post-dose 2 |  | 14–34 days post-dose 2 |  |
|---|---|---|---|---|---|
|  |  | Geometric mean ratio of (log) RocheS responses | p value | Geometric mean ratio of (log) RocheS responses | p value |
| Vaccine and schedule | AstraZeneca, 45–64 days | 0.64 (0.4–1.04) | 0.069 | 0.51 (0.34–0.77) | 0.001 |
|  | AstraZeneca, 65–84 days | 1 (ref) |  | 1 (ref) |  |
|  | AstraZeneca, 85+ days | — |  | 0.62 (0.33–1.18) | 0.148 |
|  | Pfizer, 19–29 days | — |  | 0.66 (0.45–0.97) | 0.036 |
|  | Pfizer, 65–84 days | 8.29 (5.39–12.74) | <0.001 | 6.38 (4.56–8.92) | <0.001 |
|  | Pfizer, 85+ days | 17.21 (7.16–41.38) | <0.001 | — |  |
| age group | 50–64 | 1 (ref) |  | 1 (ref) |  |
|  | 65–79 | 1.13 (0.68–1.86) | 0.634 | 0.83 (0.48–1.43) | 0.438 |
|  | 80–89 | 0.95 (0.48–1.86) | 0.877 | 1.16 (0.51–2.62) | 0.683 |
| sex | male | 1 (ref) |  | 1 (ref) |  |
|  | female | 1.68 (1.24–2.28) | 0.001 | 1.08 (0.79–1.47) | 0.019 |

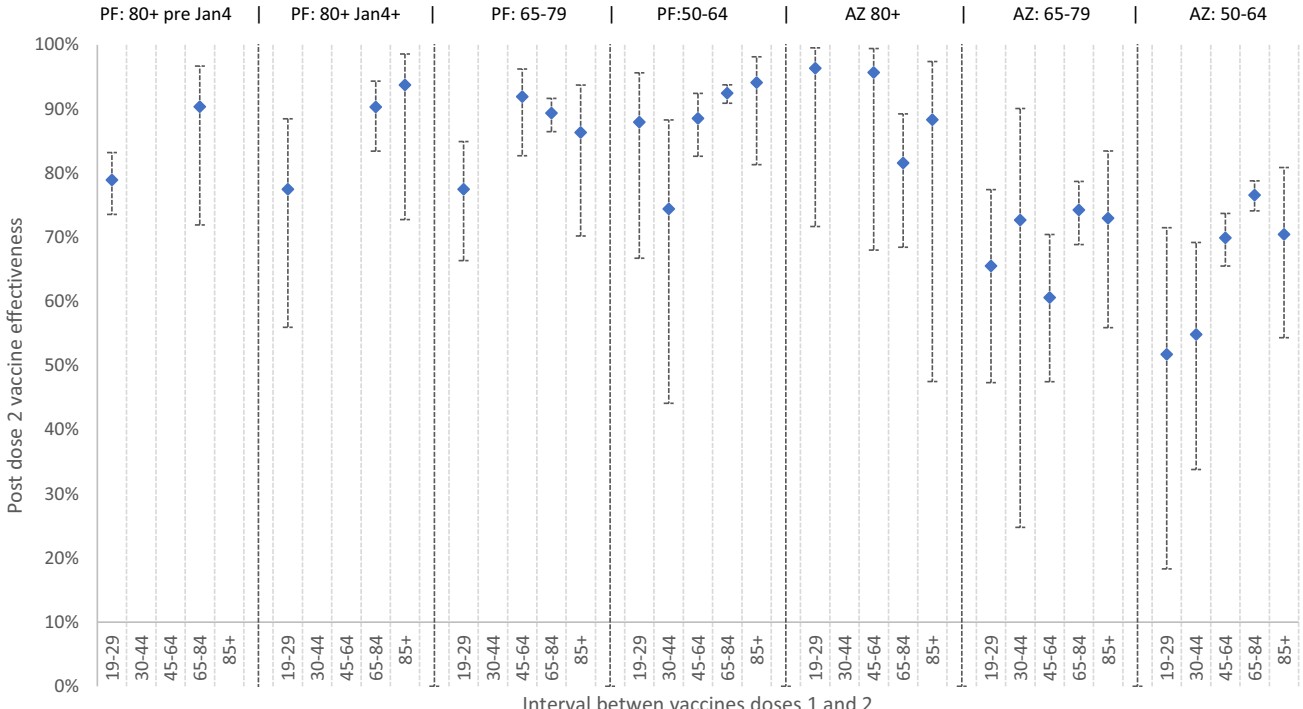

**Fig. 4 Two-dose vaccine effectiveness (with 95% CIs) by age group, vaccine type and the interval between doses.** Two-dose VE by vaccine type, the interval between doses and age group. N = 16,237 Pfizer, 20,721 AstraZeneca vaccinated people and 308,764 unvaccinated people PCR tested. Logistic regression was used to estimate the odds of vaccination in PCR-confirmed cases compared with those who tested negative for SARS-CoV-2. Vaccine effectiveness was calculated as 1 adjusted minus odds ratio and is presented as a percentage ±95% confidence interval. PF Pfizer, AZ AstraZeneca.

We were also able to compare immunogenicity with real-world VE data in England, which show substantial protection against symptomatic disease from 14 days after dose 2. Higher two-dose VE was observed with > 6-week intervals between BNT162b2 doses compared to the authorised 3-week schedule, including ≥80 year-olds. Among AZD1222 recipients, two doses provided the highest protection among ≥80 year-olds regardless of interval. Surprisingly, two-dose VE among 50–64 year-olds was lower, even when compared with adults in the same age group receiving two BNT162b2 doses. This may be due to the differential use of the vaccines in the national immunisation programme-, because AZD1222 vaccines do not require ultra-low temperatures for storage or transport, they were preferred for vaccinating care home residents who a higher risk of natural exposure and pre-existing immunity prior to vaccination, and for clinical risk groups in the community[18]. At the same time, BNT162b2 was preferentially used for healthy healthcare workers in hospitals early in the national vaccine rollout and older age groups vaccinated in the community. Since BNT162b2 was deployed earlier than AZD1222, we have longer population follow-up for this vaccine. Whilst the analyses adjust for key confounding variables, such as period and risk groups, whilst excluding those with previous COVID, it is possible that residual confounding persists to some extent. Additionally, the Alpha variant was dominant during the majority of the study period, which will affect comparisons with other international VE estimates where other variants were circulating and the follow-up time after the second dose differed[19,20]. Since May 2021, this has been replaced by the Delta variant, and whilst differences in VE between Alpha and Delta have been reported, VE remains high against hospitalisations and mortality more than 20 weeks post-vaccination[21–23]. VE estimates suggest evidence of a decline in VE beyond 10 weeks in those aged 80+ years, which could indicate immune senescence within this age group. However, large confidence intervals

make interpretation difficult and further work is required to better understand differences in VE over time across the age groups.

Notwithstanding this, it is important to emphasise that a single dose of either vaccine remains highly effective against severe endpoints, which is the primary aim of the vaccination programme, with 75–85% protection against hospitalisation in the oldest cohorts[8]. As of 28/06/2021, the vaccination programme is estimated to have prevented nearly 8 million infections and 27,000 deaths in England alone[24,25].

The strength of this study is the combination of sero-surveillance with real-world national VE data for two different vaccines in different age groups, including older adults who were excluded from initial clinical trials, with variable, real-world dosing intervals. Serological assessments provide an objective measure of vaccine responses which are important for comparing vaccines and schedules, but interpretation of serological data is limited as the way in which it correlates with protection is unknown, and the recognition that neutralising activity of antibodies and cellular immunity also play an important role in protection[26,27]. Despite this, S-antibodies have been found to correlate well with neutralising antibodies[27–29].

As with any observational study, there are limitations to VE analysis. There may be confounding factors that could increase the risk of COVID-19 in vaccinated individuals, for example, if vaccinated individuals adopted riskier behaviours after vaccination or unvaccinated individuals isolated themselves to reduce their risk of viral exposures. In addition, VE could be attenuated if there are high levels of protection from previous infection in the population or if there is misclassification of cases and test negative controls due to low sensitivity or specificity of PCR testing.

Our findings suggest higher effectiveness against infection using an extended vaccine schedule. Given the global vaccine constraints, these results are relevant to policymakers in low and

middle income countries especially in the context of highly transmissible variants and rising incidence in many parts of the world. An additional yet undervalued benefit of extended schedules is higher boosting and better protection after two doses of either vaccine, which potentially confer better protection against variants and for a longer duration than short-interval schedules. Our data also confirm previous findings of high protection after a single vaccine dose in previously-infected individuals, which is also important in the context of limited vaccine supplies. Ongoing evaluation of the protection conferred against new variants using an extended schedule will be critical.

## Method

**Participants.** CONSENSUS recruited immunocompetent adults aged ≥ 50 years, at the time of or just after vaccination, in January 2021 through London primary care networks to provide serial blood samples at 0, 3, 6, 9, 12, 15 and 20 weeks after their first dose of COVID-19 vaccine. As part of the national COVID-19 vaccine rollout, participants received either (i) two BNT162b2 doses at 3–4 weeks apart, (ii) two BNT162b2 doses up to 12 weeks apart or (iii) two AZD1222 doses up to 12 weeks apart, delivered within the community. CONSENSUS excluded people are known to be immunosuppressed and participants had to be mobile enough to attend study visits at a local vaccine centre. Antibody responses were compared with convalescent samples from adults with mild-to-moderate PCR-confirmed COVID-19, up to 98 days after symptom onset.

The sample size was determined based on providing reasonable precision for estimates of geometric mean concentrations at each timepoint, age group, the interval between doses and vaccine type for those not previously infected. To calculate precision an estimate of the standard deviation of responses post-vaccination was required. In the absence of data post-vaccination, this was estimated to be 0.5 log10 units based on data on responses post-natural infection. Using this estimate, and considering that recruitment may be variable in different groups the 95% confidence interval ±fold-widths were estimated to be ±54%, ±39% and ±26% for sample sizes of 30, 50 and 100.

The final numbers of samples available for analysis are shown in Tables 1 and 3. The smaller numbers in those aged over 80 for AstraZeneca was because this vaccine was rarely used within this age group. In addition, numbers were low for those with intervals beyond 85 days between doses because this was not primarily a group of interest when the study was designed but was selected to compare with vaccine effectiveness data.

**Serological testing.** Serum samples were tested for nucleoprotein (N) antibodies as a marker of previous SARS-CoV-2 infection (Anti-SARS-CoV-2 total antibody assay, Roche Diagnostics, Basel, Switzerland; ref: 09203079190) and spike (S) protein antibodies, which could be infection- or vaccine-derived (Elecsys Anti-SARS-CoV-2 S total antibody assay, Roche Diagnostics: positive ≥0.8 arbitrary units (au)/mL to assess vaccine response; ref: 09289275190)[30,31]. Samples were tested according to the manufacturer's instructions, with additional dilutions if necessary to get an endpoint S-antibody level.

**Assessment of vaccine effectiveness.** A test-negative case-control design was used to estimate odds ratios for testing SARS-CoV-2 positive to in vaccinated compared with unvaccinated people with COVID-19 compatible symptoms who were tested using polymerase chain reaction (PCR), as described previously[8]. The sample size was based on all available pillar 2 symptomatic cases and controls.

*Data sources*

## Outcome assessment

All healthy adults aged ≥50 years in England were eligible for inclusion. Testing for COVID-19 in the UK is done through the hospital and public health laboratories (pillar 1) and more widely through community (pillar 2) testing[32]. Pillar 2 tests performed between 26/10/2020 and 18/06/2021 we extracted for those who reported being symptomatic.

## Exposure assessment

Testing data were linked to individual vaccination histories in the National Immunisation Management System (NIMS), using unique National Health Service numbers, date of birth, surname, first name and postcode. NIMS data were extracted on 21/06/2021 with immunisation records up to 20/06/2021[33].

## Statistical analysis

*Serological assessment.* Geometric mean antibody titres (GMTs) were calculated with 95% confidence intervals (CI). Geometric mean ratios (GMR) of responses between

timepoints were estimated using a mixed regression model on log responses including a random effect for each participant, separate models were fitted for each vaccine group. The GMR of responses by vaccine type at each post-vaccination timepoint was estimated via regression on log RocheS responses and included age group and sex. Statistical analyses were performed using STATA v.14.2. Individuals testing positive on the Roche N assay were considered to have had prior SARS-CoV-2 infection. Infection status was changed if a seronegative participant seroconverted on the Roche N assay during the study and remained positive thereafter.

*Vaccine effectiveness.* Logistic regression was used to estimate the odds of vaccination in PCR-confirmed cases compared with those who tested negative for SARS-CoV-2. Only those swabbed within 0–10 days of symptom onset were included in the analysis because the sensitivity of PCR testing decreases beyond 10 days after symptom onset. Vaccine effectiveness was calculated as 1 minus odds ratio.

To estimate vaccine effectiveness in fully susceptible people, we excluded those with a previous positive PCR or antibody result prior to 8 December 2020. Estimates were adjusted for a week of onset, 5-year age bands, gender, NHS region, index of multiple deprivation (quintiles), ethnicity, health/social care worker, care home resident, flagged as clinically extremely vulnerable in NIMS, and flagged as being in the extended risk groups in NIMS among <65 year-olds only[8] Analyses were run separately for 50–64, 65–79, 80+ and 80+(early cohort) year-olds. For 50–64 year-olds, only unvaccinated or vaccinated individuals from 01/02/2021 were included because, before this, only health and social care workers were eligible in this age group. For age 65–79 and 80+ year-olds, a cut-off date of 04/01/2021 was used when the AZD1222 vaccine became available. The 80+ early cohort age group included only those unvaccinated or first dose vaccinated before 04/01/2021 with the BNT162b2 vaccine, mostly with a 3-week interval between doses.

VE was estimated by vaccine manufacturer and according to intervals after the first dose as well as from 14 days after the second dose split by intervals between first and second doses of 19–29, 30–44, 45–64, 65–84 and 85+ days.

**Reporting Summary.** Further information on research design is available in the Nature Research Reporting Summary linked to this article.

## Data availability

Applications for a relevant anonymised minimum dataset that support the findings of this study should be made to the UK Health Security Agency Office for Data Release. Source data are provided with this paper.

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

## Acknowledgements

We would like to thank Dorothy Blundell, Dr. Caroline Sayer and the team at Haverstock Healthcare GP Federation, and the whole CONSESUS team at the UKHSA including those within the Virus Reference Department at Colindale who assisted with the laboratory testing. UK Health Security Agency (Formerly known as Public Health England). The study was self-funded by Public Health England.

## Author contributions

S.N.L., F.B., P.M., M.E.R., K.E.B. and G.A. conceived and designed the CONSENSUS study; L.W. and E.L. supervised the laboratory testing; S.S., C.W., M.O.'B. and F.B. co-ordinated the patient enrolment, G.I. managed the data and H.W. performed the statistical analysis. For the VE work, J.S., C.G. and E.T. managed the VE data and N.J.A. performed the statistical analysis. G.A., J.L.B., S.N.L. and K.E.B. wrote the manuscript. All authors read and approved the submission.

## Competing interests

M.E.R. reports that the Immunisation and Countermeasures Division (UKHSA) has provided vaccine manufacturers with post-marketing surveillance reports on pneumococcal and meningococcal infection, which the companies are required to submit to the UK licensing authority in compliance with their risk management strategy. A cost-recovery charge is made for these reports. E.L. report that the PHE Vaccine Evaluation Unit does contract research on behalf of GlaxoSmithKline, Sanofi, and Pfizer, which is outside the submitted work. The remaining authors declare no competing interests.

## Ethical approval

The CONSENSUS protocol was approved by PHE Research Ethics Governance Group (reference NR0253; 18/01/2021), informed consent was obtained from participants and compensation for participation was not provided.
