## [Peer Review File · Nature Communications]

Serological responses and vaccine effectiveness demonstrate the value of extended COVID-19 vaccine schedules in EnglandEditorial Note: This manuscript has been previously reviewed at another journal that is not operating a transparent peer review scheme. This document only contains reviewer comments and rebuttal letters for versions considered at Nature Communications.

Reviewers' Comments:

Reviewer #3:

Remarks to the Author:

The authors have adequately addressed most of the comments. The study results do not allow a full appraisal of the decision to delay the second vaccine dose. Thus, study conclusions cannot support such policy (as mentioned in lines 44, 283, and 352). These statements may wrongly imply that the policy taken by most other countries with regards to timing of second dose was incorrect. Given that most authors are affiliated with Public Health England that also funded the study, conclusions should be presented in a more unequivocal and objective manner

REVIEWERS' COMMENTS

Reviewer #3 (Remarks to the Author):

The authors have adequately addressed most of the comments. The study results do not allow a full appraisal of the decision to delay the second vaccine dose. Thus, study conclusions cannot support such policy (as mentioned in lines 44, 283, and 352). These statements may wrongly imply that the policy taken by most other countries with regards to timing of second dose was incorrect. Given that most authors are affiliated with Public Health England that also funded the study, conclusions should be presented in a more unequivocal and objective manner

In light of the reviewers comments we have edited the manuscript to the following:

Abstract:

“Our findings suggest higher effectiveness against infection using an extended vaccine schedule. Given global vaccine constraints, these results are relevant to policymakers, especially with highly transmissible variants and rising incidence in many countries.”

Discussion:

“Our findings however, show sustained high levels of antibodies after the first dose to 12 weeks and is supported by the high one dose vaccine effectiveness estimates which were maintained till the second doses were given. ~~This evaluation suggests that overall the benefits of extending dosing intervals outweigh the risks from waning immunity in the UK.~~”

“Our findings suggest higher effectiveness against infection using an extended vaccine schedule. Given the global vaccine constraints, these results are relevant to policymakers in low and middle income countries especially in the context of highly transmissible variants and rising incidence in many parts of the world.”